# Anisotropy of Graphene Nanoflake Diamond Interface Frictional Properties

**DOI:** 10.3390/ma12091425

**Published:** 2019-05-01

**Authors:** Ji Zhang, Ehsan Osloub, Fatima Siddiqui, Weixiang Zhang, Tarek Ragab, Cemal Basaran

**Affiliations:** 1Department of Civil, Structural and Environmental Engineering , University at Buffalo, SUNY, Buffalo, NY 14260, USA; jzhang66@buffalo.edu (J.Z.); ehsanosl@buffalo.edu (E.O.); wzhang39@buffalo.edu (W.Z.); cjb@buffalo.edu (C.B.); 2College of Engineering, Boston University, Boston, MA 02215, USA; fatima.siddiqui8@gmail.com; 3Civil Engineering program, Arkansas State University, State University, AR 72467, USA; 4Structural Engineering Department, Alexandria University, Alexandria 21526, Egypt

**Keywords:** molecular dynamics simulation, nanomechanics, graphene nanoflake, diamond substrate, friction coefficient, interfacial mechanics

## Abstract

Using molecular dynamics (MD) simulations, the frictional properties of the interface between graphene nanoflake and single crystalline diamond substrate have been investigated. The equilibrium distance between the graphene nanoflake and the diamond substrate has been evaluated at different temperatures. This study considered the effects of temperature and relative sliding angle between graphene and diamond. The equilibrium distance between graphene and the diamond substrate was between 3.34 Å at 0 K and 3.42 Å at 600 K, and it was close to the interlayer distance of graphite which was 3.35 Å. The friction force between graphene nanoflakes and the diamond substrate exhibited periodic stick-slip motion which is similar to the friction force within a graphene–Au interface. The friction coefficient of the graphene–single crystalline diamond interface was between 0.0042 and 0.0244, depending on the sliding direction and the temperature. Generally, the friction coefficient was lowest when a graphene flake was sliding along its armchair direction and the highest when it was sliding along its zigzag direction. The friction coefficient increased by up to 20% when the temperature rose from 300 K to 600 K, hence a contribution from temperature cannot be neglected. The findings in this study validate the super-lubricity between graphene and diamond and will shed light on understanding the mechanical behavior of graphene nanodevices when using single crystalline diamond as the substrate.

## 1. Introduction

Graphene has received considerable attention during the last decade, and it is considered as a suitable material for the nanoscale electronic devices mainly because of its excellent mechanical, thermal, and electronic properties [1,2,3,4,5]. However, performance of electronic devices is significantly affected by the electrical, thermal, and mechanical properties of the device substrate [6,7]. Diamond, which is a metastable allotrope of carbon, has the potential to be used as substrate without diminishing device functions [8]. Both experiments and large-scale density functional theory (DFT) calculations have shown that when using diamond as the substrate, the most important electronic properties of graphene will be conserved, which include high current-carrying capacity [9,10]. Apart from that, another advantage is that the number of defects in the manufactured device will decrease when growing graphene nanoribbons on diamond surface [11,12]; thus, the use of graphene will benefit the quality control of graphene Nano devices. Moreover, issues related to placing graphene on metallic substrate or silicon oxide, which attenuate the electronic features, are eliminated. Diamond substrate does not influence the electronic band structure of intrinsic graphene around the Dirac point and it also improves the possibility of charge doping [13,14,15]. Considering that thermal conductivity of underlying substrates is critical for the proper functioning of all transistors [16,17], pure diamond crystal is an excellent candidate due to its efficient thermal conduction [18]. It has the highest thermal conductivity of any known material, 24–25 W cm^−1^ K^−1^ at 300 K, which is much higher compared to 1.5 W cm^−1^ K^−1^ for silicon and 4 W cm^−1^ K^−1^ for copper [18]. On the other hand, fabrication of electronic devices requires a complete evaluation of the mechanical characteristics of the substrate material, among which tribological properties are of great importance [19]. Researchers have been investigating graphite’s high lubricity characteristic and its applicability as solid lubricant [19,20,21]. 

For graphene-based electronic devices, it is critical to understand the mechanics of the graphene nanoflake–diamond interface when lateral frictional forces are present in order to design a stable nanoelectronic device. Therefore, researchers have been conducting extensive experiments using atomic force microscopy (AFM) and friction force microscopy (FFM) to explore the frictional behavior of interfaces between graphene and different types of substrate materials [6,22,23,24,25,26]. In these experiments, a diamond probing tip was sliding over the graphene surface with a certain level of normal load applied, and the friction force was measured on the diamond tip. 

In order to obtain a more realistic understanding of the interfacial friction behavior, molecular dynamics (MD) simulations can be used because of the difficulty of performing experiments at such a small length scale [20,27]. Bonelli, et al. [28] conducted studies using the tight-binding method to simulate an AFM experiment and concluded that the flake rotation angle would have a critical role in static friction magnitude. A quasi-static molecular dynamics simulation was accomplished to model the dragging of graphene nanoflakes on hexagonal boron nitride (h-BN) substrate corresponding to an FFM experiment. In their study, graphene nanoflake atoms do not have initial velocity in the sliding direction and are being pulled by the image atoms, which are sliding with a constant velocity and are connected to the graphene nanoflake via axial springs defined in the model. The authors were able to show the potential of obtaining super-lubricity at the graphene nanoflake–h-BN interface [28]. In their work, Mandelli, et al. [20] included the effect of external loads on the graphene flake, interlayer distance, and flake size. Misaligned contacts between graphene flakes and the substrate crystal is the main source of the super-lubricity. Moreover, Zhu and Li [27] investigated the sliding friction of graphene nanoflakes on gold substrate using molecular dynamics considering the effect of the size and shape of the flake and the relative rotation angle between them, known as the chirality angle [27].

Inspired by previous experiments, in this work, a comprehensive evaluation of anisotropic frictional behavior of graphene nanoflake-diamond interface has been studied by considering the effects of chirality and temperature. Based on the fact that electronic properties of graphene nanoribbon changes for different chirality, it is important to consider the entire range from armchair to zigzag graphene when studying the interface properties. Understanding the graphene flake–diamond interface properties is essential for the fabrication of graphene-based electronic transistors [29].

## 2. Molecular Dynamics Simulation Details

In order to obtain the friction coefficient of the interface between the graphene nanoflake and the diamond substrate, a molecular model of a monolayer graphene nanoflake that is sliding over a single crystal diamond substrate has been created. Molecular dynamics (MD) simulations were conducted using LAMMPS [30]. The size of the rectangular graphene nanoflake was 2 nm × 4 nm and the size of the rectangular diamond substrate was 5 nm × 10 nm, which had four layers of carbon atoms. There were 4480 atoms in the diamond substrate and 396 atoms in the armchair graphene nanoflake. In order to slide the graphene nanoflakes, atoms in the graphene nanoflake were forced to move at a constant speed of 0.1 Å/picosecond along the z direction, as shown in Figure 1. In order to allow for such sliding motion while keeping the atoms free to move under the interatomic potential, a constant velocity was applied only to the atoms in the upper and the lower boundaries. The boundaries were 5 Å wide and are shown in the green boxes in Figure 1. The graphene nanoflake edges were hydrogenated, as shown by the blue atoms in Figure 1, which was done in order to maintain the stability of the graphene and avoid the formation of sigma bonds (sp^3^-hybridization) with the diamond substrate. Therefore, van der Waals forces governed the interaction between the graphene nanoflake and the diamond substrate [31]. The atoms in the graphene nanoflake were restrained in the yz plane to maintain constant normal force over the graphene surface. The graphene nanoflake atoms between the boundaries were thermostat atoms moving with velocities that corresponded to the assigned overall temperature by the velocity scaling method [32]. These atoms did not move out of the yz plane, thus they only had velocity components in the y and z directions.

The graphene nanoflake moved over the C(100) surface of the diamond substrate, and the details of the diamond substrate are shown in Figure 2. The three Miller indices of the plane direction were 1, 0, and 0, which represent the orientation of the sliding plane with respect to the diamond face-centered cubic (FCC) lattice. In order to capture the bulk behavior of the diamond substrate, four layers of carbon atoms were simulated for computational accuracy and efficiency. The bottom layer of atoms of the diamond substrate was fixed in all three directions to provide enough stiffness for the substrate, and the top three layers of atoms were kept at a constant temperature by the velocity scaling method [32]. The simulation was first performed under absolute zero degree temperature (0 K) and then extended to 300 K, 400 K, 500 K, and 600 K. These temperatures are the most commonly selected ones for simulations of graphene electronics [17,33,34]. The time step for the MD simulations was 0.5 femtoseconds, which is less than 10% of the vibration period of a carbon atom [35] and a shrink-wrapped boundary condition was applied for the simulation box in the x, y, and z directions.

Since graphene’s properties depend significantly on the chiral angle, the graphene nanoflakes with different chiral angles, as shown in Figure 3, were simulated in order to understand the dependence of the friction coefficient on chirality. For graphene, the chiral angle between the armchair and the zigzag direction is 30°; thus, the simulations were performed with angles between these two extreme cases. In this study, graphene nanoflakes with the seven chiral angles shown in Figure 3 have been investigated. Although achiral graphene nanoflakes (zigzag and armchair) are the most thermodynamically stable orientations, it is still extremely important to simulate all possible chiralities in a friction study, because the chirality also represents the direction of the sliding along the diamond and the graphene nanoflake and not only the geometry of the graphene nanoflake itself.

Molecular dynamics simulations were performed using LAMMPS [30] in the NVT ensemble at constant temperatures of 0 K, 300 K, 400 K, 500 K, and 600 K. The atomic forces were calculated by the system, and the interactions between atoms were modeled by the adaptive intermolecular reactive empirical bond order (AIREBO) potential function given as [36]:(1)$$E=\frac{1}{2}\underset{i}{\sum }\underset{j\neq i}{\sum }\left[E_{ij}^{REBO}+E_{ij}^{LJ}+\underset{k=lj}{\sum }\underset{l\neq i,j,k}{\sum }E_{ijkl}^{TORSION}\right]$$

The definition of each term that appears in Equation (1) can be found in Zhang et al. [7]. The upper and the lower bounds of the cutoff distance was set to 1.9 Å. For the Lennard-Jones (LJ) potential, $\sigma_{CC}$ was 3.0, and thus, the cutoff distance was 10.2 Å, thus the initial distance between the graphene nanoflake and the diamond substrate is within the LJ cutoff distance. Having these cutoff distances, the strength of graphene matched the experimental results [37] and has been validated in a number of previous studies [5,7,38,39,40,41]. The interfacial friction coefficient of the graphene nanoflake–diamond substrate was calculated as the ratio between the friction force and the normal force that is applied on graphene. For each case with a specific chiral angle and temperature, the graphene nanoflake was first placed above the center of a square diamond substrate. Then it was allowed to fully relax until the thermodynamic equilibrium distance was reached, as shown in Figure 4. The initial distance between the graphene nanoflake and the diamond substrate was 7 Å, which is about twice the interlayer distance of graphite, 3.35 Å [42]. This distance is considered to be larger than the equilibrium distance. The graphene layer is considered to be fully relaxed when its distance to the diamond substrate is stable. According to observations of the behavior of the graphene layer, it stopped getting closer to the diamond substrate after just a few thousand time steps; thus, after 200,000 time steps of relaxation, graphene reached its thermodynamic equilibrium distance.

Normal force over the graphene layer was applied by decreasing the distance between graphene and diamond. After obtaining the thermodynamic equilibrium distance in each case, the normal force was almost zero, which means the graphene layer and the diamond substrate started to repel each other when having a distance less than the equilibrium distance. When sliding the graphene nanoflake along the z direction, the force needed to maintain the sliding motion in the z direction was the friction force between graphene and diamond. The variations of force in the x and z direction with respect to time were plotted, and the details of the simulation results are discussed in the following sections.

## 3. Thermodynamic Equilibrium Distance between Graphene and Diamond

The equilibrium distance between graphene and diamond was reached so that the net force on graphene fluctuated around zero. It is necessary to find the equilibrium configuration for each geometry in order to later apply normal load on the graphene in the MD simulations to then calculate the interfacial friction force. In order to find the equilibrium distance between the graphene and the diamond, the graphene nanoflake was first placed 7 Å above the center of a square 4-layer diamond substrate which was 10.0 nm × 10.0 nm, as shown in Figure 4. The relaxation was performed by fixing the bottom layer of the diamond substrate and setting all the other atoms in graphene and diamond free to move in all three directions. The relaxation stage lasted for 200,000 time steps before reaching the equilibrium configuration. The distance from the graphene layer to the diamond substrate was calculated for each atom in graphene and the probability distribution of the distance was plotted, as shown in Figure 5, for the temperature of 0 K. It can be observed that the interlayer distance closely followed a normal distribution, and the distance with the largest probability is considered as the equilibrium distance. The equilibrium distance at 0 K was used as the benchmark to be compared with other temperatures as shown in Figure 6. From the figure, the equilibrium distance between graphene and the diamond surface is around 3.35 Å, which is close to the interlayer distance of graphite [42]. The equilibrium distance increases monotonically with temperature, and this is considered to be due to more intense atomic motion at higher temperatures.

## 4. Results and Discussions

The MD simulations were performed for the interface of the rectangular graphene nanoflake of 2.0 nm × 4.0 nm sliding over the surface of a rectangular diamond substrate as shown in Figure 1. Constant normal force on the graphene layer was applied by reducing the interfacial distance from the equilibrium distance. The closest distance between graphene nanoflake and diamond substrate that has been simulated was 2.6 Å, and the corresponding total repulsive force on graphene nanoflake was around 50 nN, which is equivalent to 0.125 nN per graphene atom. Figure 7 shows the force contour in the model per atom, and it can be observed that the atomic force on graphene was distributed uniformly along its surface. Furthermore, the force on the bottom layer of the diamond substrate was much larger than the other parts of the model. Since all the atoms in the bottom layer of diamond were fixed, the net forces caused by the interactions between the atoms were eventually accumulated on the bottom layer of the substrate.

In order to understand the effects of temperature on the interfacial friction coefficient, simulations were performed at different temperatures. Firstly, simulations at absolute zero degree Kelvin were performed to serve as a reference to be compared with the results at other temperatures. When the graphene nanoflake was sliding along its armchair direction, the normal force under the distance of 2.6 Å at 0 K temperature was around 55 nN and exhibited periodicity according to Figure 8.

Friction forces were plotted with respect to the sliding distances over the diamond surface in Figure 9 for the simulation at 0 K for different chiralities. According Figure 9, the interfacial friction force between graphene and the diamond substrate exhibited stick-slip motion behavior [27] and had a periodicity of 3.6 Å. The periodicity in the frictional force is considered to be due to the periodic lattice arrangement in both graphene and diamond, and it is in agreement with the behavior of the friction force of graphene nanoflakes on Au substrate [27].

The magnitude of the fluctuation of the frictional force is taken as the frictional force of the graphene–diamond interface. The friction force versus the corresponding normal force is summarized in Figure 10. The data points of the same loading direction and simulation temperature are linearly interpolated, and the slope of the fitted line is the interfacial friction coefficient. It is observed that at 0 K, the friction coefficient varied from 0.0044 to 0.0208 for different chiralities, where the interfacial friction coefficient is the lowest along the armchair sliding direction and the highest along the zigzag direction. The interfacial friction coefficients of the graphene–diamond interface are generally lower than those of the graphene–gold interface [27].

As temperature may have a significant influence on the friction coefficient, the MD simulations were extended to 300 K, 400 K, 500 K, and 600 K. Figure 11 shows the interfacial frictional force against the sliding distance for different rotation angles at 300 K. Similar to the results at 0 K temperature, periodicity was observed in the 300 K simulation results. The friction force of each loading direction was plotted with respect to the corresponding normal force in order to calculate the interfacial friction coefficient. As shown in Figure 11, the stick-slip motion was more evident when changing the sliding direction from armchair to zigzag direction. The interfacial frictional force was developed in the stick step and then dropped during the slip process.

In Figure 12, the interfacial friction coefficients of different angles of chirality at different temperatures are compared. The first observation is that the friction coefficients at 0 K always have the lowest value for all the rotation angles. There was no lattice vibration at 0 K; thus, the contribution from thermal fluctuations does not exist. 

The friction coefficient between graphene and diamond was within the range of ~0.004 to ~0.023 at 300 K depending on chirality. For all the cases that were simulated, the friction coefficients were within ~0.004 to ~0.024. The interfacial friction coefficient always had the smallest value when the angle between graphene and the diamond was 0 degrees, which is the armchair direction of the graphene nanoflake. In comparison, the interfacial friction coefficient has the largest value when the sliding direction is along the zigzag direction of the graphene nanoflake, which has a 30 degrees angle between graphene and the diamond. The interfacial friction coefficients of all the simulations accompanied with the calculated friction angles for each case are summarized in Table 1. It is clearly visible from the table that increasing the temperature from 300 K to 600 K led to an increase of around 20% in the friction coefficient for the armchair chirality. 

Contrary to the findings of Dienwiebel and Frenken [43], we found that as the temperature increases, the friction coefficient increases. To explain the effect of the temperature on the friction coefficient, it is important to realize that as the interatomic distance between the diamond substrate and the graphene nanoflake decreases, the interatomic interaction will increase. Greater force will be required to overcome this interaction, which leads to a higher friction coefficient. As explained earlier in the paper, the two layers are brought to a distance from one another that is less than the equilibrium interatomic distance at the simulated temperature, thus generating a repulsive force. This distance is the averaged distance of the fluctuation due to the thermal energy. It is expected that as the temperature increases, the interatomic distances between the atoms within the graphene nanoflake or the diamond substrate increase, and thus the interlayer distance decreases, leading to the higher friction coefficients between graphene and diamond at higher temperatures. It is important to note that the temperature increase will have the same effect on the long-range interaction between the two layers modeled using the Lennard Jones potential and could thus cancel the effect of temperature on the interaction between the atoms within the same layer. But this did not happen due to the fact that the forces of the AIREBO potential were significantly higher than the forces of the Lennard-Jones potential. As a result, the two competing effects increase the friction coefficient as the temperature increases.

The friction coefficients of the graphene–diamond interface were compared with relevant previous studies, as shown in Figure 13. Berman, et al. [44] studied the lubrication effect of the interface between diamond-like carbon (DLK) and graphene, nanodiamond, and graphene-wrapped nanodiamond, respectively. They found the coefficient of friction is the lowest for the interface that is lubricated by graphene-wrapped nanodiamond, which is ~0.004 ± 0.002. The interface with graphene only has a friction coefficient of ~0.04 ± 0.01 and of only ~0.07 ± 0.01 with nanodiamond particles. A similar experiment was performed by Bhowmick, et al. [45], which investigated the friction properties between multilayer graphene (MLG) and different counterface materials. For example, they compared the friction coefficient of MLG sliding against the surface of hydrogenated diamond-like carbon (DLC) and N-based coated steel. The N-based coatings included TiN, TiAlN, TiCN, and CrN. The friction coefficient of graphene-hydrogenated diamondlike carbon (h-DLC) interface was ~0.08±0.01. For the interface of graphene-uncoated 52100 steel and graphene–TiCN interface, the friction coefficient was 0.15 ± 0.03 and 0.26 ± 0.03, respectively. Generally, the friction coefficient of graphene sliding along the armchair direction was the lowest, and it was lower than the graphene–h-DLC, graphene–uncoated 52100 steel, and graphene–TiCN interfaces. The friction coefficient of zigzag graphene–diamond interface was the highest, and it was comparable with that of the graphene–TiCN interface

The increasing friction coefficient from armchair direction to zigzag direction is considered to be due to more intense interactions between graphene and the diamond substrate. In order to verify this hypothesis, we introduce the term Proximity (P) to describe the intensity of the interaction between the graphene flake and the diamond substrate as the summation of the reciprocal of the pairwise distance (P=∑i,j1li,j) between atoms of the graphene nanoflake (i) and atoms in the diamond substrate (j). This represents a quantitative value for the adhesion energy but is easier to calculate as it can be related to the energy of the standard 12/6 Lennard-Jones potential energy E which is $4\varepsilon \left[{\left(\frac{\sigma }{r}\right)}^{12}-{\left(\frac{\sigma }{r}\right)}^{6}\right]$. This expression is used as the benchmark of the interaction of two groups of atoms. For instance, when the graphene layer is getting close to the diamond substrate, it is expected to have more atomic interactions in the system. More atoms will be within the interaction range, and the reciprocal of the pairwise distance increases; thus, the summation of the reciprocal of the pairwise distance will increase. On the contrary, when the graphene layer is getting far away from the diamond substrate, it is expected to have less interaction between graphene and diamond substrate. Both the number of atomic pairs of atoms within the interaction range and the reciprocal of the distance between the atoms will decrease, and eventually, the summation of the reciprocal of the pairwise distance decreases.

A MATLAB code was developed to calculate the Proximity between graphene nanoflake atoms and the atoms in the diamond substrate, and only the pairs that were within the Lennard-Jones cutoff distance of 10.2 Å were taken into account. The Proximity was calculated with respect to the time steps of the relaxation for both armchair and zigzag graphene nanoflake. The results indicate a decrease from armchair to zigzag graphene nanoflake from 4120 1/Å to 3690 1/Å, which means after relaxation, the zigzag graphene nanoflake is expelled farther from the diamond substrate compared to the armchair graphene nanoflake. Thus, it is expected that there is more atomic interaction for the zigzag graphene nanoflake compared to armchair graphene nanoflake at the same interlayer distance. The observation complies with the increase in the interfacial friction coefficient from the armchair direction to the zigzag direction obtained from the MD simulation results summarized in Table 1.

## 5. Conclusions

Molecular dynamics simulations have been performed for graphene nanoflake sliding on the C(100) surface of a single crystalline diamond substrate. Relaxation simulations were first performed in order to find the equilibrium distance between graphene and diamond at different temperatures. The equilibrium distance between graphene and diamond was close to 3.35 Å, which is the interlayer distance of graphite. 

The friction forces exhibited periodic stick-slip motion with respect to the sliding distance, and it was due to the periodic lattice arrangement of both graphene and diamond. The simulation was first performed under 0 K and then extended to other temperatures—300 K, 400 K, 500 K, and 600 K. The friction coefficient was the lowest when graphene was sliding along its armchair direction and the highest when it was sliding along the zigzag direction. Moreover, the interfacial friction coefficient can increase by 20% with a rise in temperature from 300 K to 600 K. In summary, the friction coefficient ranged from 0.0042 to 0.0244 depending on the chirality angle and the temperature. Compared to the graphene–N-coated 52100 steel interface, the friction coefficient of the armchair graphene–diamond interface was lower, while the friction coefficient of the zigzag graphene–diamond interface was comparable. The above findings can help to understand the behavior of graphene-based nanodevices sitting on single crystalline diamond substrate. 

## Figures and Tables

**Figure 1 materials-12-01425-f001:**
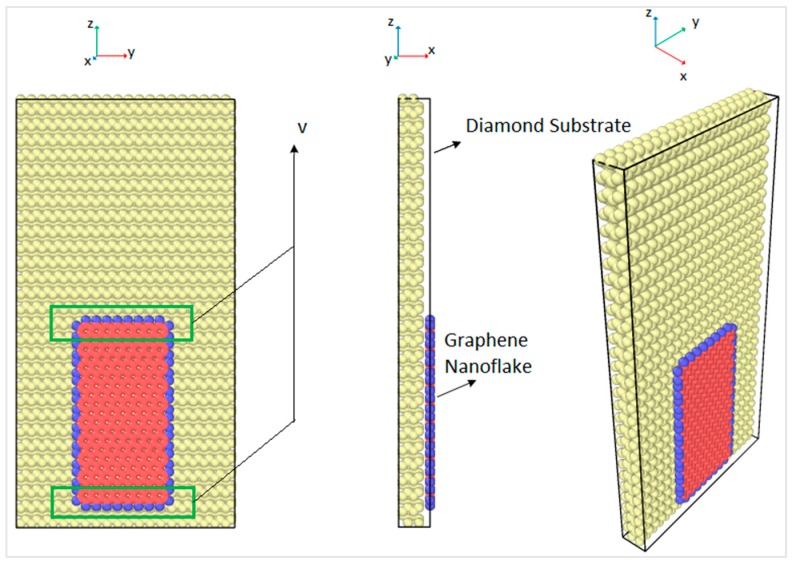
Molecular dynamics model and the loading scheme.

**Figure 2 materials-12-01425-f002:**
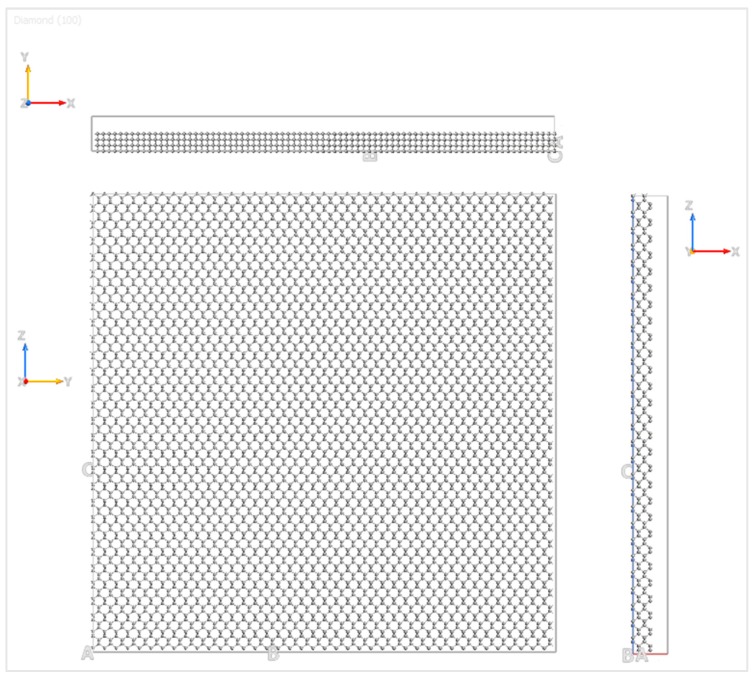
Spatial orientation of the diamond substrate.

**Figure 3 materials-12-01425-f003:**
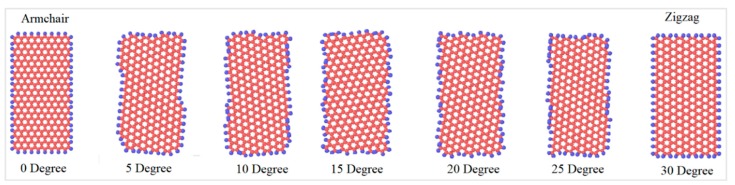
Simulated graphene nanoflakes with different chiral angles with respect to the substrate.

**Figure 4 materials-12-01425-f004:**
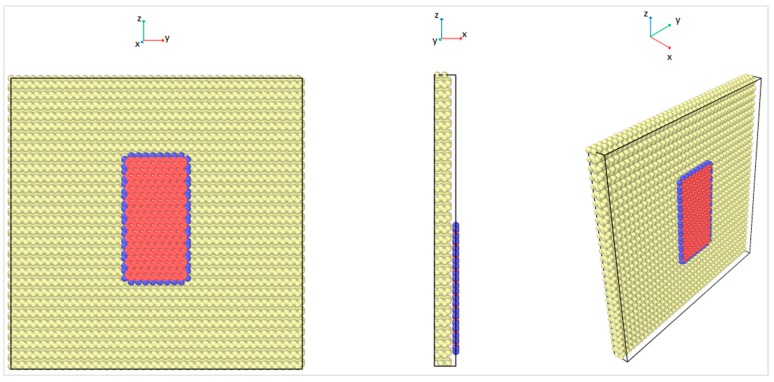
Geometry of the relaxation simulation.

**Figure 5 materials-12-01425-f005:**
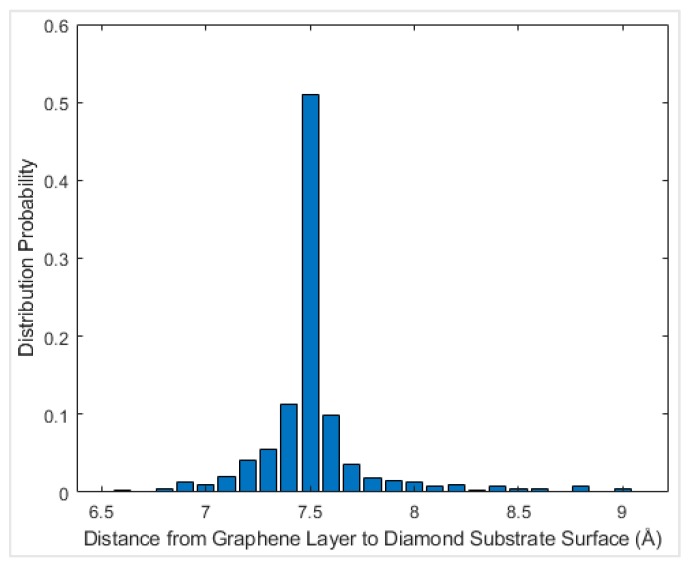
Probability distribution of the distance between graphene nanoflake and the diamond surface at 0 K.

**Figure 6 materials-12-01425-f006:**
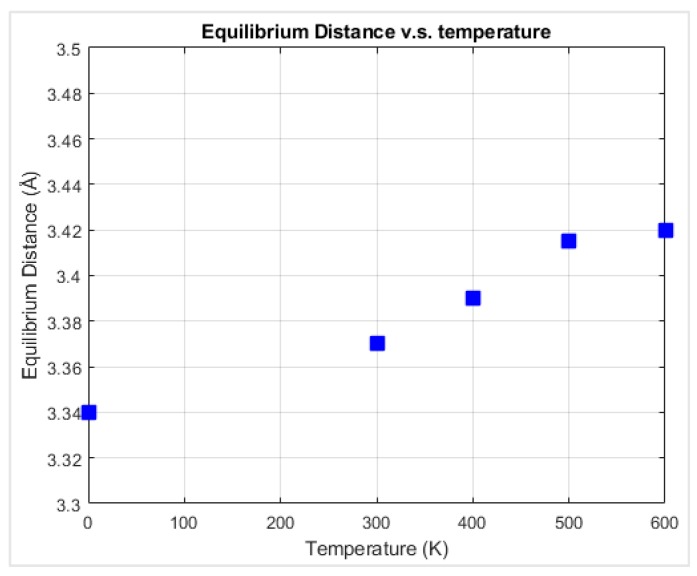
Thermodynamic equilibrium interlayer distance between the diamond substrate and the graphene nanoflake versus temperature.

**Figure 7 materials-12-01425-f007:**
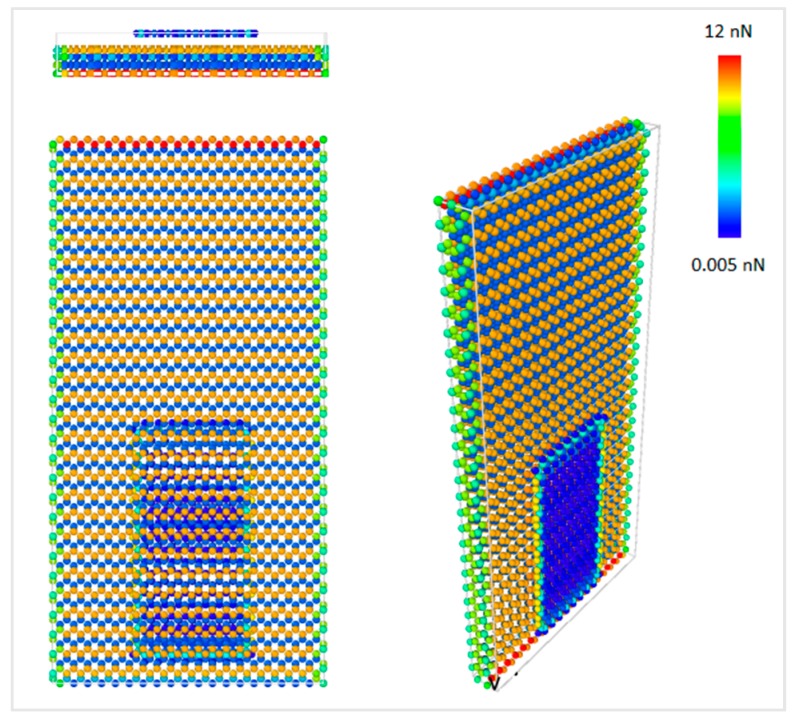
Atomic force contour of graphene during sliding with 0 degree chirality at 300 K.

**Figure 8 materials-12-01425-f008:**
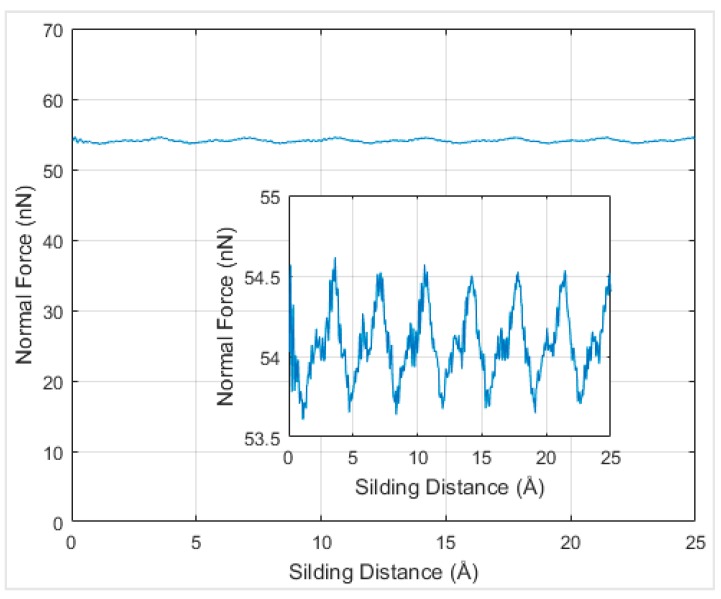
Normal force versus the sliding distance for armchair chirality under 2.6 Å distance at 0 K temperature.

**Figure 9 materials-12-01425-f009:**
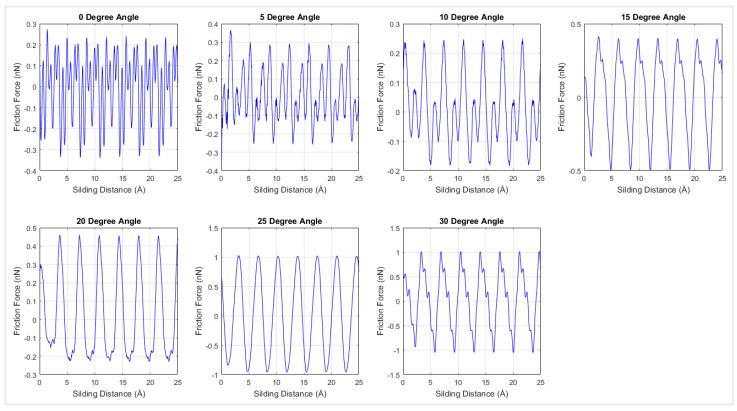
Friction force versus the sliding distance for different chiralities under 2.6 Å distance at 0 K temperature.

**Figure 10 materials-12-01425-f010:**
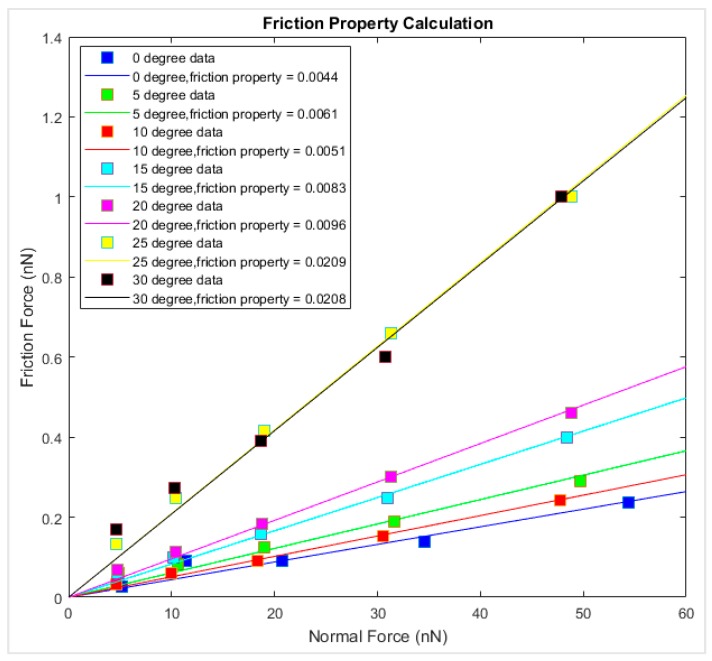
Friction force versus normal force for different chiralities at 0 K.

**Figure 11 materials-12-01425-f011:**
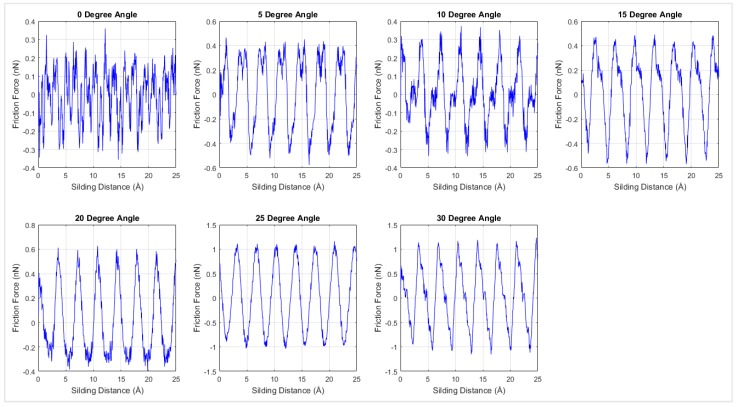
Interfacial friction force versus sliding distance at 300 K temperature for different loading directions.

**Figure 12 materials-12-01425-f012:**
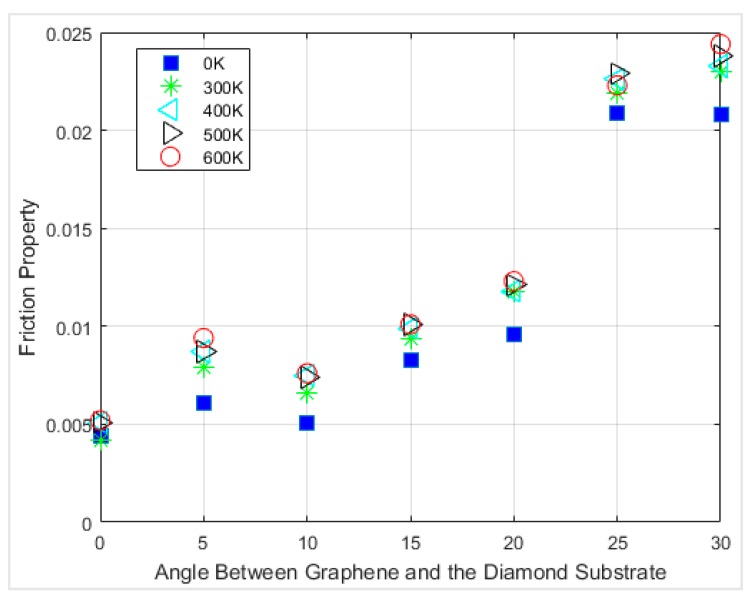
Interfacial friction coefficient versus sliding direction at different temperatures.

**Figure 13 materials-12-01425-f013:**
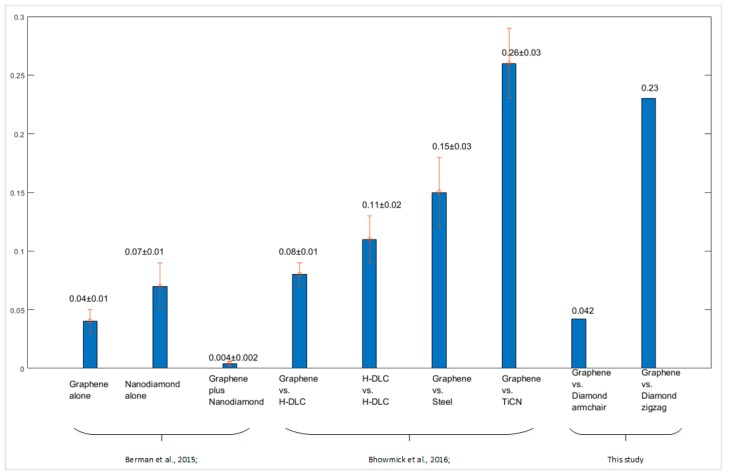
Comparison with previous studies by Berman, et al. [44] and Bhowmick, et al. [45] on the friction properties of different interfaces.

**Table 1 materials-12-01425-t001:** Interfacial friction coefficient of graphene nanoflake on diamond substrate.

Temperature	0 K		300 K		400 K		500 K		600 K	
Rotation Angle	Interfacial Friction Coefficient	Friction Angle (Degree)	Interfacial Friction Coefficient	Friction Angle (Degree)	Interfacial Friction Coefficient	Friction Angle (Degree)	Interfacial Friction Coefficient	Friction Angle (Degree)	Interfacial Friction Coefficient	Friction Angle (Degree)
**0**	0.0044	0.25	0.0042	0.24	0.0051	0.29	0.0051	0.29	0.0052	0.30
**5**	0.0061	0.35	0.0079	0.45	0.0087	0.50	0.0087	0.50	0.0094	0.54
**10**	0.0051	0.29	0.0066	0.38	0.0075	0.43	0.0074	0.42	0.0076	0.44
**15**	0.0083	0.48	0.0094	0.54	0.0099	0.57	0.0101	0.58	0.0101	0.58
**20**	0.0096	0.55	0.0118	0.68	0.0118	0.68	0.0121	0.69	0.0123	0.70
**25**	0.0209	1.20	0.0219	1.25	0.0226	1.29	0.0229	1.31	0.0223	1.28
**30**	0.0208	1.19	0.023	1.32	0.0233	1.33	0.0238	1.36	0.0244	1.40

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
