# Peer review of "Anisotropy of Graphene Nanoflake Diamond Interface Frictional Properties"

_materials, 2019, doi:10.3390/ma12091425_

Round 1
Reviewer 1 Report
The authors studied the friction of a nano-flake of graphene over a diamond surface using classical molecular dynamics.
The introduction should be improved, in particular the graphene properties on the first sentence should be rephrased. The reference number 1 is missing and the refs. 2-17 are too many at the same time and too many of the same authors. It would be better to add less references (eliminating some of refs from 5 to 17) and insert a review on the graphene properties and features.
The sentence ..."key electronic advantages of graphene..." is not clear.
In section 2: why authors used a 0.1A/ps force? where they found this number? Add a reference or justify this choice.
Figures 1, 3 and 4 should be re-colored. Both the substrate and the graphene are colored in red and readers have difficulties in understanding the differences. I suggest to authors to change the colors of both diamond and graphene improving the understanding of the figures.
Why authors used the temperatures 300K, 400K, 500K and 600K? add a reference or justify this choice.
Figure 2 should be merged with Figure 1 or Figure 3
In figure 4 it seems that the diamond substrate is bigger than in Figure 1. Is it correct? Why? Do authors changed the size of the system?
The sentence "The calculation of Proximity from the MATLAB code..." is not clear and should be rephrased.
In the discussion, authors compare their results only with the friction over gold. I suggest to broaden the discussion adding comparisons with other cases, such as the work
Friction reduction mechanisms in multilayer graphene sliding against hydrogenated diamond-like carbon of Bhowmick et al.
or
Macroscale Superlubricity enabled by graphene nanoscoll formation of Berman et al.
Author Response
Response to the reviewer’s comments:
The authors studied the friction of a nano-flake of graphene over a diamond surface using classical molecular dynamics.
The introduction should be improved, in particular the graphene properties on the first sentence should be rephrased. The reference number 1 is missing and the refs. 2-17 are too many at the same time and too many of the same authors. It would be better to add less references (eliminating some of refs from 5 to 17) and insert a review on the graphene properties and features.
According to the reviewer’s comments, the first sentence has been rephrased as “Graphene has received considerable attention during the last decade and it is considered as a suitable material for the nano-scale electronic devices mainly because of its excellent mechanical, thermal and electronic properties.”.
The reference number 1 was in the abstract and it has been deleted and cited in the introduction and results. Some of the references in 2-17 have been deleted and only the most related ones remained. Also, the graphene properties feature related to the topic of this study have been reviewed in the following part of the introduction.
The sentence ..."key electronic advantages of graphene..." is not clear.
According to the reviewer’s comment, the sentence has been rephrased as “Both experiments and large scale density functional theory (DFT) calculations have shown that when using diamond as the substrate, the most important electronic properties of graphene will be conserved, which include high current-carrying capacity”.
In section 2: why authors used a 0.1A/ps force? where they found this number? Add a reference or justify this choice.
This is the loading speed we assumed in order to get the results of the friction coefficient of graphene-diamond interface. It can be any number and the loading speed may have an influence on the friction coefficient.
[1-4].
Figures 1, 3 and 4 should be re-colored. Both the substrate and the graphene are colored in red and readers have difficulties in understanding the differences. I suggest to authors to change the colors of both diamond and graphene improving the understanding of the figures.
According to reviewer’s comments, the color of the diamond substrate in Figures 1, 3, and 4 are changed so that readers can separate the graphene flake from the diamond substrate.
Why authors used the temperatures 300K, 400K, 500K and 600K? add a reference or justify this choice.
The temperatures we selected are the most commonly selected ones for simulation graphene electronics. Previous studies using similar temperature range are:
Chu, Yanbiao, Tarek Ragab, and Cemal Basaran. "Temperature dependence of Joule heating in zigzag graphene nanoribbon." Carbon 89 (2015): 169-175. [5]
Ragab, Tarek, and Cemal Basaran. "Joule heating in single-walled carbon nanotubes." Journal of Applied Physics 106.6 (2009): 063705. [6]
Ragab, Tarek, and Cemal Basaran. "Semi-classical transport for predicting joule heating in carbon nanotubes." Physics Letters A 374.24 (2010): 2475-2479. [7]
We have added the above references in the manuscript.
Figure 2 should be merged with Figure 1 or Figure 3.
In figure 4 it seems that the diamond substrate is bigger than in Figure 1. Is it correct? Why? Do authors changed the size of the system?
We used different diamond geometry for the relaxation and loading, i.e., the shape of the diamond substrate for relaxation is square and large in size in order to let the graphene flake rotate freely without the limitation of the diamond size. Figure 2 is used specifically for showing the atomic structure of the diamond substrate and Figure 3 is showing a zoom out of the relaxation geometry and thus does not clearly show the atomic structure.
The sentence "The calculation of Proximity from the MATLAB code..." is not clear and should be rephrased.
According to the comment of the reviewer, the sentence has been rephrased as “The results indicate a decrease from armchair to zigzag graphene nano-flake from 4120 1/Å to 3690 1/Å, which means after relaxation zigzag graphene nano-flake is expelled farther from the diamond substrate compared to armchair graphene nano-flake.”.
In the discussion, authors compare their results only with the friction over gold. I suggest to broaden the discussion adding comparisons with other cases, such as the work
Friction reduction mechanisms in multilayer graphene sliding against hydrogenated diamond-like carbon of Bhowmick et al.
or
Macroscale Superlubricity enabled by graphene nanoscoll formation of Berman et al.
The friction coefficients of the graphene-diamond interface have been compared with previous relevant studies as shown in Figure 1. Berman, et al. [8] studied the lubrication effect of the interface between diamondlike carbon (DLK) and graphene, nanodiamond, and graphene-wrapped nanodiamond, respectively. They found the coefficient of friction is the lowest for the interface that is lubricated by graphene-wrapped nanodiamond, which is ~0.004±0.002. The interface with graphene only has friction coefficient of ~0.04±0.01 and ~0.07±0.01 with nanodiamond particles only. Similar experient has been performed by Bhowmick, et al. [9], which is about the friction properties between multilayer graphene (MLG) and different counterface materials. For example, they compared the friction coefficient of MLG sliding against the surface of hydrogenated diamond-like carbon (DLC) and N-based coated steel. The N-based coatings include TiN TiAlN, TiCN and CrN. The friction coefficient of graphene-hygrogenated diamondlike carbon (h-DLC) interface is ~0.08±0.01. For the interface of graphene-uncoated 52100 Steel and graphene-TiCN interface, the friction coefficient is 0.15±0.03 and 0.26±0.03, respectively. Generally, the friction coefficient of graphene sliding along the armchair direction is the lowest and it is lower than the graphene-h-DLC, graphene-uncoated 52100 Steel, and graphene-TiCN interfaces. The friction coefficient of zigzag graphene-diamond interface is the highest and it is comparable with that of the graphene-TiCN interface.
Figure 1 Comparison with previous studies of Berman, et al. [8] and Bhowmick, et al. [9]
References:
[1] J. Zhang, T. Ragab, and C. Basaran, "Influence of vacancy defects on the damage mechanics of graphene nanoribbons," International Journal of Damage Mechanics, vol. 26, no. 1, pp. 29-49, 2017.
[2] J. Zhang, T. Ragab, and C. Basaran, "The effects of vacancy defect on the fracture behaviors of zigzag graphene nanoribbons," International Journal of Damage Mechanics, vol. 26, no. 4, pp. 608-630, 2017.
[3] J. Zhang, T. Ragab, and C. Basaran, "Comparison of fracture behavior of defective armchair and zigzag graphene nanoribbons," International Journal of Damage Mechanics, vol. 28, no. 3, pp. 325-345, 2019.
[4] J. Zhang, W. Zhang, T. Ragab, and C. Basaran, "Mechanical and electronic properties of graphene nanomesh heterojunctions," Computational Materials Science, vol. 153, pp. 64-72, 2018/10/01/ 2018.
[5] Y. Chu, T. Ragab, and C. Basaran, "Temperature dependence of Joule heating in Zigzag Graphene Nanoribbon," Carbon, vol. 89, pp. 169-175, 2015/08/01/ 2015.
[6] T. Ragab and C. Basaran, "Joule heating in single-walled carbon nanotubes," Journal of Applied Physics, vol. 106, no. 6, p. 063705, 2009.
[7] T. Ragab and C. Basaran, "Semi-classical transport for predicting joule heating in carbon nanotubes," Physics Letters A, vol. 374, no. 24, pp. 2475-2479, 2010.
[8] D. Berman, S. A. Deshmukh, S. K. Sankaranarayanan, A. Erdemir, and A. V. Sumant, "Macroscale superlubricity enabled by graphene nanoscroll formation," Science, vol. 348, no. 6239, pp. 1118-1122, 2015.
[9] S. Bhowmick, A. Banerji, and A. Alpas, "Friction reduction mechanisms in multilayer graphene sliding against hydrogenated diamond-like carbon," Carbon, vol. 109, pp. 795-804, 2016.

Reviewer 2 Report
This paper investigated the anisotropic frictional behavior of graphene nano-flake-diamond interface by performing MD simulations in which a graphene nano-flake slides on (100) surface of a single crystalline diamond. They mainly considered two factors of chirality and temperature: for chirality, the friction coefficient is the lowest when graphene is sliding along its armchair direction and the highest when sliding along the zigzag direction; for temperature, the interfacial friction coefficient can increase by 20% with rising of the temperature from 300 K to 600 K. Thus, this paper is helpful to the fabrication engineering of graphene-based electronic transistors.
However, two issues are still needed to work on:
1.In figure 1 and 4, the colors of both diamond substrate and graphene flake are the same, which make readers difficult to distinguish the two kinds of materials. A better representation is needed.
2. The authors mentioned that the diamond substrate surface is (100), but why they chose this surface? Does it have special meanings for the real application? Or other slip system should be considered and discussed.
Author Response
Response to the reviewer’s comments:
This paper investigated the anisotropic frictional behavior of graphene nano-flake-diamond interface by performing MD simulations in which a graphene nano-flake slides on (100) surface of a single crystalline diamond. They mainly considered two factors of chirality and temperature: for chirality, the friction coefficient is the lowest when graphene is sliding along its armchair direction and the highest when sliding along the zigzag direction; for temperature, the interfacial friction coefficient can increase by 20% with rising of the temperature from 300 K to 600 K. Thus, this paper is helpful to the fabrication engineering of graphene-based electronic transistors.
However, two issues are still needed to work on:
1.In figure 1 and 4, the colors of both diamond substrate and graphene flake are the same, which make readers difficult to distinguish the two kinds of materials. A better representation is needed.
According to the reviewer’s comment, we have changed the color of the substrate in Figure 1 and Figure 4 so that readers can distinguish the graphene flake from the diamond substrate.
2. The authors mentioned that the diamond substrate surface is (100), but why they chose this surface? Does it have special meanings for the real application? Or other slip system should be considered and discussed.
The selection of C(100) surface plane for the substrate was done to allow for comparison with previous work done on the frictional properties of graphene over Au substrate [1]. Same surface was chosen as the plane orientation for the diamond planes. Also, in the literature[1], C(100) is the surface orientation which complies with the experimental studies.
[1] P. Zhu and R. Li, "Study of Nanoscale Friction Behaviors of Graphene on Gold Substrates Using Molecular Dynamics," Nanoscale Research Letters, vol. 13, no. 1, p. 34, 2018/02/02 2018.

Reviewer 3 Report
paper “Anisotropy of Graphene Nano-flake Diamond Interface Frictional Properties” by Ji Zhang et al. studies the frictional properties of the interface between graphene nano-flake and single crystalline diamond, considering the effect of the relative sliding angle, and of temperature.
The authors find that when considering graphene nanoflakes with different chiral angles, the friction coefficient varies. Their study validates superlubricity effects, being the friction coefficient dependent on the sliding direction.
The authors also find that the friction coefficient increases when increasing temperature. No previous experimental work regarding thermolubricity is cited (see for instance Dienwiebel and Frenkel “Experimental observations of superlubricity and thermolubricity”, Chapter 8 in Gnecco, E. (editor) Fundamentals of Friction and Wear 2nd Edition; Springer (2015)). And the authors statement in pg. 9 line 135, that the increase of the intensity of the thermal vibrations of the atoms leads to higher interactions forces between atoms in the graphene nano-flake and the diamond substrate, does not seem necessarily true to this reviewer.
Hence, I suggest that the authors revise the part of the paper related to the temperature dependence of the friction coefficient and discuss their work in the context of previous similar experimental and theoretical studies.
Author Response
Response to the reviewer’s comments:
paper “Anisotropy of Graphene Nano-flake Diamond Interface Frictional Properties” by Ji Zhang et al. studies the frictional properties of the interface between graphene nano-flake and single crystalline diamond, considering the effect of the relative sliding angle, and of temperature.
The authors find that when considering graphene nanoflakes with different chiral angles, the friction coefficient varies. Their study validates superlubricity effects, being the friction coefficient dependent on the sliding direction.
The authors also find that the friction coefficient increases when increasing temperature. No previous experimental work regarding thermolubricity is cited (see for instance Dienwiebel and Frenkel “Experimental observations of superlubricity and thermolubricity”, Chapter 8 in Gnecco, E. (editor) Fundamentals of Friction and Wear 2nd Edition; Springer (2015)). And the authors statement in pg. 9 line 135, that the increase of the intensity of the thermal vibrations of the atoms leads to higher interactions forces between atoms in the graphene nano-flake and the diamond substrate, does not seem necessarily true to this reviewer.
Hence, I suggest that the authors revise the part of the paper related to the temperature dependence of the friction coefficient and discuss their work in the context of previous similar experimental and theoretical studies.
According to the reviewer, we have referred the “Experimental observations of superlubricity and thermolubricity”, Chapter 8 in Gnecco, E. (editor) Fundamentals of Friction and Wear 2nd Edition; Springer (2015)) in our study.
We have also added the comment below in our manuscript:
“Contrary to the findings of Dienwiebel and Frenken [1], we found that as the temperature increases the friction coefficient increases. To explain the effect of the temperature on the friction coefficient, it is important to realize that as the interatomic distance between the diamond substrate and the graphene nano-flake decreases, the interatomic interaction will increase. Greater force will be required to overcome this interaction, which leads to higher friction coefficient. As explained earlier in the paper the two layers are brought to some distance from each other that is less than the equilibrium interatomic distance at the simulated temperature and thus generating a repulsive force. This distance is the averaged distance of the fluctuation due to the thermal energy. It is expected that as the temperature increases the interatomic distances between the atoms within the graphene nano-flake or the diamond substrate increase and thus the interlayer distance decreases. It will lead to the higher friction coefficients between the graphene and diamond at higher temperatures. It is important to note that the temperature increase will have the same effect on the long range interaction between the two layers modeled using the Lennard Jones potential and thus could cancel the effect of the temperature on the interaction between the atoms within the same layer. But this did not happen due to the fact that the forces of the AIREBO potential is significantly higher than the forces of the Lennard-Jones potential. As a result, the two competing effects will increase the friction coefficient when the temperature increases.”.
Reference
[1] M. Dienwiebel and J. W. M. Frenken, "Experimental Observations of Superlubricity and Thermolubricity," in Fundamentals of Friction and Wear on the Nanoscale, E. Gnecco and E. Meyer, Eds. Cham: Springer International Publishing, 2015, pp. 139-156.

Round 2
Reviewer 1 Report
Changes in the paper have been done in according to comments and suggestions of reviewers.
The paper in the present form is easier to read.